# Association between Alcohol Consumption and Metabolic Dysfunction-Associated Steatotic Liver Disease Based on Alcohol Flushing Response in Men: The Korea National Health and Nutrition Examination Survey 2019–2021

**DOI:** 10.3390/nu15183901

**Published:** 2023-09-07

**Authors:** Dae Eon Kang, Si Nae Oh

**Affiliations:** 1Department of Family Medicine, Severance Hospital, Seoul 03722, Republic of Korea; dek93@yuhs.ac; 2Department of Family Medicine, National Health Insurance Service Ilsan Hospital, Goyang 10444, Republic of Korea; 3Department of Medicine, Yonsei University Graduate School, Seoul 03722, Republic of Korea

**Keywords:** metabolic dysfunction-associated steatotic liver disease (MASLD) alcohol consumption, alcohol flushing, lipid accumulation product, Korean men, KNHANES

## Abstract

Metabolic dysfunction-associated steatotic liver disease (MASLD) is distinguished by the buildup of excessive liver fat unrelated to alcohol consumption. However, the role of alcohol consumption on disease progression is debatable. Recently, alcohol flushing syndrome in Asian populations has gained interest, and its role in the risk of developing MASLD is unknown. Therefore, in this cross-sectional study, we investigated the association between alcohol consumption and MASLD in Korean men, considering their alcohol flushing response and utilizing the lipid accumulation product (LAP) score. Data from the Korean National Health and Nutrition Examination Survey (2019–2021) were analyzed. Participants were categorized into non-or-infrequent drinkers and light-to-heavy drinkers and further sub-classified based on alcohol flushing response as non-flushers and flushers. Multivariate logistic regression analysis showed a significant association between alcohol consumption and MASLD risk in both non-flushers (aHR 1.90, 95% CI 1.51–2.40, *p* < 0.001) and flushers (aHR 2.35, 95% CI 1.94–2.84, *p* < 0.001) after adjusting for potential confounding factors such as age, exercise, smoking, body mass index, systolic blood pressure, total cholesterol, and fasting plasma glucose. There was a significant interaction between alcohol consumption and alcohol flushing response for MASLD risk (*p* for interaction < 0.001). These findings emphasize the importance of alcohol flushing as a potential indicator of MASLD risk in Korean men and highlight the need for further research to understand the underlying mechanisms and develop targeted preventive strategies.

## 1. Introduction

Metabolic dysfunction-associated steatotic liver disease (MASLD) is a widespread and worrisome liver condition characterized by excessive accumulation of fat in the liver that is not caused by alcohol consumption [1,2]. It has become a significant global health concern affecting numerous individuals worldwide [1]. Although MASLD development is influenced by various factors, the effect of alcohol consumption in its progression continues to be controversial and requires investigation [2].

Recently, the genetic aspects of alcohol flushing syndrome have garnered attention, especially among Asian populations, including Koreans. Alcohol flushing syndrome, commonly referred to as the “Asian flush”, is a physiological reaction marked by facial flushing, rapid heart rate, and nausea after consuming alcohol [3]. This response is linked to a deficiency in the enzyme alcohol dehydrogenase, which is vital for alcohol metabolism, resulting in the buildup of acetaldehyde, a toxic byproduct during alcohol breakdown [4].

The prevalence of alcohol flushing syndrome is significantly higher among Asians, owing to genetic variations in alcohol-metabolizing enzymes, particularly in genes encoding alcohol dehydrogenase (ADH) and aldehyde dehydrogenase (ALDH) [5,6]. Among these genetic variations, the ALDH2*2 allele, commonly found in Asian populations, is associated with reduced enzyme activity, leading to acetaldehyde accumulation and contributing to alcohol-flushing responses [7,8,9]. Although the relationship between alcohol consumption and MASLD has been extensively studied, the influence of alcohol flushing syndrome and its role in causing MASLD remains relatively unexplored, particularly in Korean men. Therefore, evaluating the impact of alcohol flushing response on the risk of developing MASLD is crucial, as it may offer valuable insights into individual susceptibility to liver disease, especially in populations with a high prevalence of alcohol flushing syndrome, such as Koreans. Additionally, the identification of specific risk factors may pave the way for targeted preventive strategies to alleviate the burden of MASLD in high-risk populations.

The purpose of this study was to examine the association between alcohol consumption and MASLD in Korean men using the lipid accumulation product (LAP) score as a noninvasive marker of MASLD risk while considering their facial flushing response [10,11,12,13]. We hypothesized that the alcohol flushing response, influenced by genetic variations in alcohol-metabolizing enzymes, may modify the relationship between alcohol consumption and MASLD risk, leading to different outcomes in individuals with and without flushing [14]. Data from the Korean National Health and Nutrition Examination Survey (KNHANES) conducted between 2019 and 2021 were utilized [15,16]. This research endeavor seeks to decipher the intricate interplay between alcohol consumption, genetic predisposition (alcohol flushing syndrome), and the risk of MASLD in Korean men. By examining these multifaceted factors within a single framework, we aspire to provide a holistic understanding of MASLD etiology, tailored health interventions, and, ultimately, improved outcomes for high-risk individuals.

## 2. Materials and Methods

### 2.1. Study Participants

This cross-sectional study collected data from the eighth edition of the KNHANES, conducted between January 2019 and December 2021. The selection process of study participants is shown in Figure 1. Initially, the dataset comprised of 10,354 men. To ensure the relevance of the sample, stringent exclusion criteria were implemented, which led to the exclusion of individuals outside the age range of 19–64 years, resulting in 6074 eligible participants. Furthermore, participants with pre-existing health conditions such as type 2 diabetes mellitus, hepatitis B (HBV), hepatitis C (HCV), or liver cirrhosis were also excluded, leaving a final sample number of 5300 participants [17,18,19]. Individuals with missing data necessary for the analysis were excluded to enhance the data quality and minimize potential bias. Ultimately, a representative cohort of 5134 Korean men from the KNHANES (2019–2021) was used to investigate the relationship between alcohol consumption, alcohol flushing response, and risk of developing MASLD.

### 2.2. Alcohol Flushing Response

Study participants’ alcohol-flushing responses were assessed using a questionnaire. Those who indicated that they ‘often’ or ‘always’ experienced facial flushing after alcohol consumption were classified as flushers. Participants who answered ‘no’ to the same question were categorized as non-flushers, indicating the absence of a facial flushing response following alcohol consumption.

### 2.3. Alcohol Consumption

The alcohol consumption level of the study participants was evaluated through a questionnaire. Non-or-infrequent drinkers were identified as individuals who answered, ‘I have not drunk at all in the past year’, ‘less than once a month’, or ‘about once a month’ in the questionnaire [20,21]. In contrast, light-to-heavy drinkers included all participants who answered, “I drank more than once a month”, indicating that they had consumed alcohol at a higher frequency during the past year [22,23,24].

### 2.4. Metabolic Dysfunction-Associated Steatotic Liver Disease

The main outcome variable was the presence or absence of MASLD. Liver biopsy is the preferred method for diagnosing MASLD and liver cirrhosis [13]. Nonetheless, this approach is invasive and unsuitable for follow-up assessments. Alternative diagnostic techniques for MASLD include ultrasonography, which is less accurate when detecting mild steatosis, and magnetic resonance imaging proton density fat fraction (MRI-PDFF), which offers greater precision but comes at a higher cost and is only available in limited quantities [13]. Due to the restriction of the administrative data, we used the LAP score as a diagnostic marker for MASLD, which indicates excessive fat accumulation in the liver unrelated to alcohol consumption [25,26,27]. 

LAP score is an indicator used to estimate excessive lipid buildup [13]. It is calculated using waist circumference (WC) and fasting plasma triglycerides (TG) as follows: (LAP  =  (WC (cm) − 65) × TG (mmol/L)) for men [13]. An LAP score cut-off of 46.43 was used as an indicator of MASLD risk [16]. Additionally, we incorporated the hepatic steatosis index (HSI), which takes into account BMI, diabetes status, ALT, and AST. This index utilizes a cut-off value of 36 to evaluate the likelihood of hepatic steatosis [28]. Moreover, the NAFLD liver fat score (NLFS) was designed to indicate the probability of fibrosis, factors in age, hyperglycemia, BMI, platelet count, albumin levels, and AST/ALT ratio. The NLFS employs a cut-off of −0.64 to make assessments [29].

### 2.5. Covariates

Several covariates were considered in the analysis to account for potential confounding factors, including age, exercise level, smoking status, body mass index (BMI), systolic blood pressure, total cholesterol levels, and fasting plasma glucose levels.

The participants’ weight and height were measured while wearing light clothing and no shoes. Body mass index (BMI) was calculated by dividing weight in kilograms by the square of height in meters. Blood pressure (BP) was measured using a sphygmomanometer following the standard manual instructions. Glucose levels were assessed using the hexokinase ultraviolet method. Cholesterol levels, which included triglyceride (TG) and high-density lipoprotein cholesterol (HDL-C) levels, were determined using standard enzymatic methods [30,31,32,33]. All measurements were performed in a certified central laboratory using an automatic analyzer (Hitachi Ltd., Tokyo, Japan) after the participants had fasted for a minimum of 8 h.

In the questionnaire, the participants’ smoking habits were categorized into three groups based on their responses related to their lifetime and current smoking status: non-smokers, ex-smokers, and current smokers. The participants’ exercise habits were determined based on their responses to questions on high, moderate, and low-intensity physical activity during leisure time and were categorized as non-exercisers, regular exercisers, or irregular exercisers.

### 2.6. Statistical Analysis

Statistical analyses were performed to examine the associations and differences between various variables. For continuous variables such as age, BMI, blood pressure, cholesterol, ALT, AST, fasting plasma glucose, and LAP score, the *t*-test was used to compare non-or-infrequent drinkers and light-to-heavy drinkers, as well as non-flushers and flushers, assuming a normal distribution. However, for the LAP score, which exhibited a non-normal distribution, the non-parametric–Mann–Whitney U test was applied. For categorical variables such as smoking and exercise, the chi-square test was used to assess the associations between these variables and alcohol consumption. Multivariate logistic regression analysis was used to estimate the adjusted odds ratios (aORs) and 95% confidence intervals (CIs) for MASLD separately for non-flushers and flushers. We conducted tests to examine the interactions between the alcohol consumption group and the alcohol flushing response group in relation to MASLD risk. Statistical significance was set at *p* < 0.05. Data management and analyses were performed using IBM SPSS statistics.

### 2.7. Ethics Consideration

All participants of the Korea National Health and Nutrition Examination Survey provided written informed consent. The study protocol for the survey was approved by the Institutional Review Board of the Korea Centers for Disease Control and Prevention. The approval number is 2018-01-03-5C-A.

## 3. Results

### 3.1. Characteristics of Study Participants

Table 1 displays the characteristics of the study participants. The study encompassed a total of 5134 participants categorized into non-or-infrequent drinkers (*n* = 1827) and light-to-heavy drinkers (*n* = 3307), further stratified by flushing response: non-flushers (n = 1270) and flushers (*n* = 557) among non-or-infrequent drinkers, and non-flushers (*n* = 997) and flushers (*n* = 2310) among light-to-heavy drinkers. Comparative analysis revealed subtle distinctions in age, with light-to-heavy drinkers being younger (39.08 years, SD = 13.22) compared to non-or-infrequent drinkers (43.42 years, SD = 13.33). High-density lipoprotein (HDL) cholesterol exhibited noteworthy variation, with light-to-heavy non-flushers having the highest mean (50.4 mg/dL, SD = 11.81) and non-or-infrequent flushers the lowest (46.07 mg/dL, SD = 10.55). The HSI (hepatic steatosis index) score showed significance among groups, with light-to-heavy non-flushers having the highest mean (34.26, SD = 6.84), as did the LAP (lipid accumulation product) score, with non-or-infrequent drinkers having the lowest mean (38.83, SD = 49.5). Additionally, alcohol consumption demonstrated significant differences between all drinking and flushing subgroups (*p* < 0.001 *). Smoking and exercise patterns also differed significantly among the groups (*p* < 0.001 * and *p* = 0.03 *, respectively). In summary, the study delineated comprehensive participant characteristics, highlighting subtle variations in age, cholesterol levels, health scores, alcohol consumption, smoking, and exercise patterns across distinct drinking and flushing cohorts.

### 3.2. Association between Alcohol Consumption and MASLD Risk by LAP

Table 2 displays the adjusted odds ratios (aORs) for MASLD, cut-offed by LAP, based on alcohol consumption among both non-flushers and flushers as determined by multivariate analysis. The age-adjusted odds ratio for MASLD in the light-to-heavy drinkers group for non-flushers was 1.22 (95% CI: 1.03–1.44, *p* = 0.020) compared to non-or-infrequent drinkers. However, after adjusting for multiple variables, including age, exercise, smoking, body mass index, systolic blood pressure, total cholesterol, and fasting plasma glucose, the aOR increased to 1.90 (95% CI: 1.51–2.40, *p* < 0.001). This indicated a significantly elevated risk of developing MASLD on alcohol consumption.

In flushers, the age-adjusted OR for MASLD in moderate–to–heavy drinkers was 1.47 (95% CI: 1.29–1.68, *p* < 0.001) compared with non-or-infrequent drinkers. After multivariable adjustment, the aOR further increased to 2.35 (95% CI: 1.94–2.84, *p* < 0.001), indicating a stronger association between alcohol consumption and MASLD risk in this group (*p* for interaction < 0.001).

### 3.3. Association between Alcohol Consumption and MASLD Risk by HSI

Table 3 presents the aORs and corresponding 95% CIs for MASLD, stratified by the HSI, based on alcohol consumption among non-flushers and flushers. In the age-adjusted analysis, individuals categorized as non-or-infrequent drinkers were used as the reference group, showing a statistically significant association with MASLD risk in both non-flushers (aOR: 1.06, 95% CI: 1.02–1.10, *p* = 0.003) and flushers (aOR: 1.07, 95% CI: 1.04–1.10, *p* < 0.001). After comprehensive adjustment for confounding factors, including age, smoking, exercise, body mass index, systolic blood pressure, total cholesterol, and fasting plasma glucose, the association remained significant in both non-flushers (aOR: 1.05, 95% CI: 1.01–1.09, *p* = 0.011) and flushers (aOR: 1.07, 95% CI: 1.04–1.11, *p* < 0.001). The interaction between alcohol consumption and HSI was not statistically significant (*p* = 0.118), suggesting that the impact of alcohol consumption on MASLD risk was consistent across different levels of hepatic steatosis as determined by HSI.

### 3.4. Association between Alcohol Consumption and MASLD Risk by NFLS

In Table 4, odds ratios (aORs) and their corresponding 95% confidence intervals (CIs) are presented to explore the relationship between MASLD risk, defined by NLFS, and alcohol consumption among both non-flushers and flushers. The age-adjusted analysis revealed that compared to non-or-infrequent drinkers, individuals classified as light-to-heavy drinkers exhibited a slightly elevated risk of MASLD among both non-flushers (aOR: 1.05, 95% CI: 1.02–1.09, *p* = 0.003) and flushers (aOR: 1.05, 95% CI: 1.02–1.08, *p* < 0.001). Upon considering multiple covariates such as age, smoking, exercise, body mass index, systolic blood pressure, total cholesterol, and fasting plasma glucose, the association remained significant in both non-flushers (aOR: 1.04, 95% CI: 1.01–1.08, *p* = 0.011) and flushers (aOR: 1.03, 95% CI: 1.01–1.07, *p* = 0.006). The interaction analysis did not yield a statistically significant result (*p* = 0.102), suggesting that the interplay between alcohol consumption and NLFS-defined MASLD risk did not differ significantly between non-flushers and flushers. Abbreviations used include aOR for adjusted odds ratio and CI for confidence interval. Adjustment for various factors, including age, smoking, exercise, body mass index, systolic blood pressure, total cholesterol, and fasting plasma glucose, was incorporated into the multivariable analysis.

## 4. Discussion

In this study, we revealed a notable association between alcohol consumption and MASLD risk with respect to alcohol flushing syndrome. Light-to-heavy alcohol consumption was associated with a 1.9-fold higher risk of MASLD in non-flushers and a 2.35-fold higher risk in flushers. Our statistical test for interactions indicated that the alcohol-flushing response had a significant effect on the relationship between alcohol consumption and MASLD risk. These findings suggest that individuals with alcohol flushing syndrome may be more vulnerable to the detrimental effects of alcohol on the liver, resulting in a more pronounced impact on MASLD risk than in non-flushers. Our study adds novel insights by highlighting the role of alcohol flushing as a potential indicator of MASLD risk in Korean men.

MASLD is a significant global health concern, and its association with alcohol consumption is debated [34]. Previous studies have reported an increased risk of MASLD with alcohol consumption [35]. In addition to our findings regarding MASLD and alcohol consumption, it is worth noting the emergence of a new category known as MetALD, which represents individuals with both metabolic dysfunction-associated steatotic liver disease (MASLD) and increased alcohol intake [36]. While our study primarily focuses on the association between alcohol consumption and MASLD among Korean men, the introduction of MetALD as a distinct category holds significance in the broader context of liver disease research [36]. MetALD highlights a subgroup of individuals who not only exhibit metabolic dysfunction associated with liver steatosis but also engage in elevated alcohol consumption [36]. Given the limitations in the KNHANES dataset, we were unable to extract specific data for MetALD patients. Nonetheless, the concept of MetALD underscores the need for a nuanced understanding of liver diseases, particularly in the context of alcohol-related liver disease [36]. This new categorization emphasizes that MASLD may coexist with increased alcohol intake, potentially influencing the disease’s course and management. Future research endeavors should aim to investigate the prevalence, risk factors, and tailored clinical approaches for individuals with MetALD. Our study contributes to the broader discourse on liver health by highlighting the importance of considering both metabolic factors and alcohol intake, recognizing that the interplay between these factors may have implications for disease susceptibility and progression. Exploring the unique characteristics and clinical implications of MetALD will be essential to better inform patient care and public health strategies.

The results of our investigation compellingly establish a noteworthy connection between alcohol consumption and the susceptibility to MASLD among Korean men. This pivotal discovery not only reinforces the existing body of knowledge but also enriches it by delving into the specific characteristics of alcohol flushing response as an influencing factor. Both individuals classified as non-flushers and those categorized as flushers exhibited a heightened risk of developing MASLD with increased alcohol consumption. This finding reinforces the significance of considering both the frequency and quantity of alcohol intake as critical determinants in assessing MASLD risk. Moreover, it underlines the importance of accounting for individual variations in alcohol metabolism, of which flushing response is a prominent indicator.

This intriguing phenomenon suggests that alcohol flushing response could serve as a valuable clinical marker for identifying individuals at a heightened risk of MASLD. Incorporating this physiological parameter into risk assessment strategies could potentially enhance the accuracy of predicting MASLD susceptibility and guide targeted preventive measures. Implementing preventive strategies and targeted interventions tailored to individuals with alcohol flushing syndrome could prove valuable in reducing the burden of MASLD. Moreover, if healthcare professionals are aware of the potential liver-related risks associated with alcohol consumption in flushers, they can emphasize the importance of alcohol abstinence or moderation in this subgroup. Additionally, the organization of public health campaigns could also play a role in educating the population about the link between the alcohol flushing response and liver health, encouraging individuals to monitor their alcohol consumption patterns based on their flushing response. These insights not only enhance our understanding of MASLD’s etiology but also provide a glimpse into the potential future of healthcare, an era where preventive strategies and treatment approaches are tailored to individual genetic profiles. The integration of genetics into the management of MASLD could revolutionize how we approach liver health, from risk assessment to targeted interventions. As researchers continue to peel back the layers of complexity in the interplay between genetics, alcohol, and liver disease, we edge closer to a future where healthcare truly becomes personalized, offering each individual the best chance for optimal liver health based on their unique genetic characteristics.

The underlying mechanisms that explain the differential association between alcohol consumption and MASLD risk in flushers and non-flushers warrant further investigation. Alcohol flushing syndrome manifests as facial flushing and other physiological responses after alcohol intake owing to reduced enzyme activity, leading to acetaldehyde accumulation [37]. 

The exacerbation of MASLD risk in individuals with alcohol flushing syndrome could be linked to variations in alcohol metabolism pathways that trigger cascades of events within hepatocytes. Beyond the immediate discomfort of flushing, the compromised breakdown of acetaldehyde might lead to a higher accumulation of toxic byproducts, which in turn could contribute to oxidative stress and inflammation, key drivers of MASLD [13]. It is plausible that the impaired alcohol metabolism and increased acetaldehyde accumulation in flushers exacerbate hepatic lipid accumulation and oxidative stress, contributing to the heightened risk of MASLD in this subgroup [13].

The intricate relationship between ALDH2 (aldehyde dehydrogenase 2) and metabolic dysfunction-associated steatotic liver disease (MASLD) adds a layer of complexity to our understanding of this prevalent metabolic liver disorder. Global epidemiological trends highlight the escalating incidence of metabolic liver diseases, with MASLD encompassing a wide spectrum of liver pathologies [38]. The progressive stages of MASLD have been found to intersect with alcohol-metabolizing enzymes and alcohol metabolism processes [38]. Research by Li et al. revealed notable alterations in alcohol-related enzymes and oxidative stress markers in MASLD progression, shedding light on the potential impact of ALDH2 in this context [39]. Notably, ALDH2, a key player in alcohol metabolism, showed significant changes in both mRNA and protein levels in various stages of MASLD. While ALDH2 protein levels increased in metabolic-associated steatohepatitis (NASH) groups, ALDH1A1 and ALDH1B1 protein levels were reduced. This intricate interplay hints at the potential role of ALDH2 in MASLD pathogenesis [39].

Central to these findings is the intriguing role of 4-hydroxy-nonenal (4-HNE), a product of oxidative stress, in the regulation of ALDH2 activity [40]. The covalent modification of ALDH2 by 4-HNE poses the possibility of inhibiting ALDH2 function, thus impacting alcohol metabolism [40]. The protective effects of Alda-1, which enhances ALDH2 detoxification activity and mitigates hepatic lipid content, hint at potential interventions that could target ALDH2-related dysregulation in MASLD [40]. The intricate web of alcohol-metabolizing enzymes, including ADH and other ALDH isoforms, as well as CYP2E1 and CAT, further underscores the multifaceted nature of alcohol’s interactions within the context of MASLD [40]. The higher expression of these enzymes in MASLD livers reveals the potential mechanistic importance of alcohol metabolism in this disorder.

The intricate association between ALDH2 and metabolic dysfunction-associated steatotic liver disease (MASLD) introduces a complex interplay that extends beyond alcohol metabolism, touching on key processes contributing to liver inflammation and cellular damage. An integral aspect of this interaction is the alcohol flushing response, a prominent phenomenon observed in individuals with ALDH2 deficiency. This reaction arises due to the accumulation of acetaldehyde, an inherently toxic byproduct of alcohol metabolism that remains unconjugated due to impaired ALDH2 activity [41]. The implications of the alcohol flushing response reach beyond discomfort, potentially shaping the landscape of liver inflammation and cellular injury observed in MASLD [42].

Central to this dynamic is acetaldehyde, a molecule of considerable reactivity that can inflict direct cellular damage by forming adducts with proteins and DNA, thereby impairing their function and integrity [42]. In the context of MASLD, the cumulative oxidative stress stemming from the dysregulation of redox homeostasis further exacerbates acetaldehyde-induced damage [43]. Reactive oxygen species (ROS) generated during oxidative stress contribute to a damaging environment characterized by increased lipid peroxidation and oxidative damage to cellular components [43].

The amalgamation of acetaldehyde accumulation and oxidative stress sets the stage for a potent inflammatory milieu within the liver [44]. Acetaldehyde, known to upregulate pro-inflammatory cytokines and activate immune cells, can synergize with the already heightened inflammation characteristic of MASLD [44]. Chronic inflammation, a hallmark of MASLD pathogenesis, is orchestrated by an intricate interplay of immune cells and cytokines that, if left unchecked, can promote tissue damage and fibrosis [44]. The culmination of these processes not only magnifies cellular injury but also fosters a cycle of sustained inflammation that exacerbates MASLD progression [45].

The pivotal role of ALDH2 and the alcohol flushing response in this cascade is becoming increasingly apparent. By contributing to acetaldehyde accumulation and amplifying oxidative stress, ALDH2 deficiency intertwines with the complex web of metabolic disturbances and inflammatory signals driving MASLD. This interplay illuminates a previously underappreciated facet of MASLD progression, emphasizing that the impact of ALDH2 extends beyond alcohol metabolism.

However, the relationship between alcohol consumption and MASLD remains a topic of debate. Emerging evidence suggests a nuanced relationship between moderate alcohol consumption and MASLD progression. Studies such as the National Health and Nutrition Examination Survey imply the potential benefits of modest alcohol consumption, linking it to decreased mortality among MASLD patients. Yet, contrasting results from large-scale cohort studies, like the one conducted in Korea, have raised concerns about the impact of even moderate alcohol consumption on MASLD progression. As such, the interaction between ALDH2 polymorphism and MASLD, especially in the context of alcohol consumption, remains a subject ripe for further investigation. Altogether, these findings point to the intricate interplay between ALDH2, alcohol consumption, and MASLD, urging further research to uncover the potential therapeutic avenues and lifestyle interventions in MASLD management.

Exacerbation of MASLD can give rise to a spectrum of undesirable consequences, ranging from mild to severe [46]. MASLD entails the accumulation of surplus fat within liver cells and can evolve through distinct stages, each carrying its own potential repercussions. As MASLD advances, it can transform into a more serious variant termed metabolic-associated steatohepatitis (NASH) [47]. NASH is marked by inflammation and harm to liver cells [47]. This progression may lead to the creation of scar tissue (fibrosis) in the liver, which, over time, can culminate in cirrhosis [48]. Cirrhosis denotes the advanced scarring of liver tissue that hampers liver function [48]. This can give rise to complications like portal hypertension, ascites (accumulation of abdominal fluid), and hepatic encephalopathy (brain dysfunction due to liver failure) [49]. In the advanced phases of MASLD, especially cirrhosis, the risk of hepatocellular carcinoma, a form of liver cancer, is elevated [50]. Ultimately, severe cirrhosis can result in liver failure, necessitating a liver transplant for survival [50].

Furthermore, MASLD is frequently linked with cardiovascular risk factors like obesity, diabetes, and dyslipidemia [51]. The exacerbation of MASLD can heighten the susceptibility to cardiovascular events such as heart attacks and strokes [51]. The connection between MASLD and insulin resistance often brings about an augmented risk of developing or exacerbating type 2 diabetes as MASLD worsens [52]. The deterioration of MASLD can contribute to the progression of metabolic syndrome, a combination of obesity, high blood pressure, elevated blood sugar, and abnormal cholesterol levels [48]. This syndrome amplifies the risk of cardiovascular disease and type 2 diabetes. Our study raises intriguing questions about the broader metabolic implications of alcohol flushing syndrome. Research exploring the relationship between alcohol flushing and conditions such as obesity, insulin resistance, and diabetes might uncover additional aspects of this phenotype’s impact on health. Investigating these broader metabolic effects could illuminate the intricate web of physiological responses influenced by genetic variations in alcohol metabolism pathways.

Although this study yields valuable insights, it has several limitations that deserve acknowledgment. First, the cross-sectional design prevented us from establishing causal relationships between alcohol consumption, alcohol flushing response, and MASLD risk. Longitudinal studies are necessary to confirm the temporal sequence of these associations. Second, the reliance on self-reported alcohol consumption data introduces the possibility of recall and social desirability biases. The accuracy of alcohol consumption estimates can be improved by incorporating objective measures, such as biomarkers or alcohol metabolite levels. Third, the study population consisted solely of Korean men, limiting the generalizability of the findings to other ethnicities and sexes. The replication of this study in diverse populations is essential to assess the broader impact of alcohol flushing on the risk of MASLD. Fourthly, despite taking several confounding factors into consideration in this study, the potential role of unmeasured or residual confounding factors influencing the observed association remains. Other lifestyle factors, dietary habits, and genetic variations that were not considered in our analysis could have potentially affected the results [53]. Fifthly, the use of the LAP, HSI, and NLFS score as a noninvasive marker for MASLD risk has inherent limitations, as it may not accurately capture all cases of MASLD [54]. Although a liver biopsy remains the gold standard for diagnosing MASLD, its impracticality in large-scale studies necessitates the use of alternative measures [55]. 

Furthermore, it is essential to recognize the limitations specific to our consideration of MetALD (metabolic dysfunction-associated alcoholic liver disease) in this study [36]. While we introduced MetALD as a meaningful category based on the definition of MASLD with increased alcohol intake, our analysis was constrained by the dataset’s limitations. This limitation hinders our ability to provide a detailed characterization of MetALD patients, their prevalence, and the distinct risk factors associated with this subgroup. Despite these limitations, our study serves as a crucial starting point in acknowledging the existence and relevance of MetALD, paving the way for more in-depth exploration of this category in future research. Another limitation is the absence of genetic variation data among the study participants. Genetic factors, including polymorphisms of key enzymes such as ALDH2, can exert a substantial influence on individual responses to alcohol consumption and its subsequent impact on MASLD risk. The ALDH2 gene, for instance, carries polymorphisms that result in varying enzyme activity levels, affecting alcohol metabolism efficiency and thus potentially modifying MASLD susceptibility. Given the lack of genetic data in our study, we were unable to explore the potential interplay between ALDH2 polymorphism, alcohol flushing response, and MASLD risk among Korean men. Future research endeavors with comprehensive genetic data could offer valuable insights into the intricate genetic and environmental interactions driving MASLD pathogenesis and its association with alcohol consumption patterns. Lastly, the constrained ability to precisely calculate alcohol intake due to the limitations of the KNHANES dataset was an important limitation of this study. The available data primarily provided categorical information on alcohol consumption patterns, such as frequency and types of beverages consumed, rather than specific quantities. Despite these limitations, our study provided insights into the broader relationship between alcohol consumption, genetic factors, and MASLD risk in Korean men. Future research should address these limitations to provide a more nuanced understanding of the complex interplay between alcohol flushing response, genetics, and alcohol consumption in the context of MASLD susceptibility.

Despite these limitations, our study highlights the significance of considering the alcohol-flushing response when evaluating the impact of alcohol consumption on the risk of MASLD. However, future studies should address these limitations and delve deeper into the underlying mechanisms to develop targeted interventions and preventive measures to reduce the burden of MASLD in vulnerable populations.

## 5. Conclusions

In summary, this cross-sectional study provides a comprehensive exploration of the relationship between alcohol consumption and the risk of MASLD in the male population of Korea. Our study not only substantiates the link between alcohol consumption and MASLD risk among Korean men but also accentuates the critical role of alcohol flushing response in shaping this association. By recognizing the heightened vulnerability of this subgroup, healthcare providers can design and implement strategies that resonate with their specific needs and challenges. These interventions may encompass educational initiatives to raise awareness about the intricate interplay between alcohol consumption, flushing response, and MASLD risk. Additionally, lifestyle interventions and counseling programs could empower individuals to make informed choices regarding alcohol consumption and adopt behaviors that mitigate MASLD risk. Further large longitudinal studies are necessary to confirm the temporal sequence of these associations.

## Figures and Tables

**Figure 1 nutrients-15-03901-f001:**
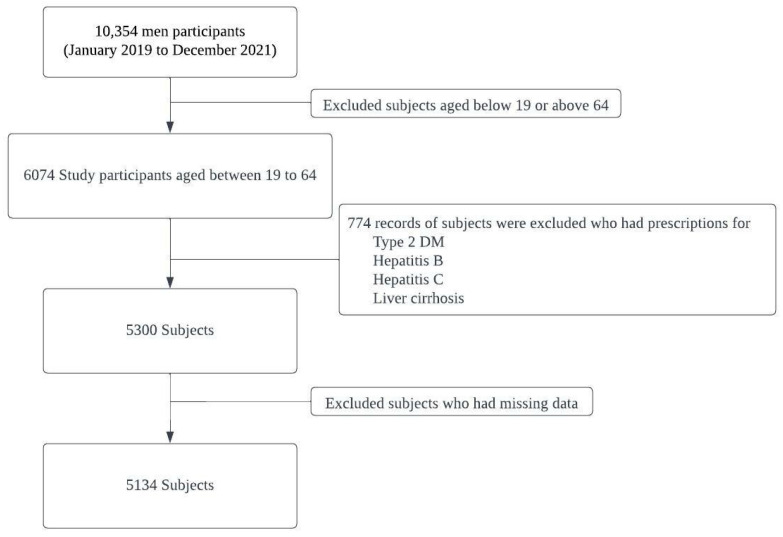
Selection process of study participants.

**Table 1 nutrients-15-03901-t001:** Characteristics of the study participants.

Characteristics	Total (*n* = 5134)	Non-or-Infrequent Drinker (*n* = 1827)		Light-to-Heavy Drinker (*n* = 3307)		*p*-Value
		Non-Flusher (*n* = 1270)	Flusher (*n* = 557)	Non-Flusher (*n* = 997)	Flusher (*n* = 2310)	
Age (y)	42.92 (13)	43.42 (13.33)	39.08 (13.22)	43.35 (12.88)	43.48 (12.66)	0.01
Body mass index (kg/m^2^)	24.93 (3.61)	24.77 (3.68)	24.79 (4.14)	24.87 (3.45)	25.07 (3.48)	0.46
Systolic blood pressure (mmHg)	119.57 (13.44)	117.3 (12.81)	117.09 (13.18)	119.82 (13.48)	121.32 (13.55)	0.06
Diastolic blood pressure (mmHg)	78.82 (9.91)	76.91 (9.45)	77.31 (10.08)	78.86 (9.74)	80.21 (9.96)	<0.001
Total cholesterol (mg/dL)	195.89 (36.46)	193.98 (36.08)	193.12 (36.07)	195.61 (35.43)	197.75 (37.1)	0.32
Triglyceride (mg/dL)	159.94 (140.6)	140.87 (118.43)	141.1 (101.66)	161.44 (144.85)	174.32 (155.86)	0.09
High-density lipoprotein cholesterol (mg/dL)	48.72 (11.42)	46.07 (10.55)	47.09 (10.63)	49.04 (11.28)	50.4 (11.81)	0.01
Low-density lipoprotein cholesterol (mg/dL)	115.18 (36.15)	119.73 (34.75)	117.82 (32.95)	114.28 (36.03)	112.48 (37.45)	0.41
Fasting plasma glucose (mg/dL)	99.82 (19.14)	98.05 (17.83)	98.07 (23.03)	99.77 (17.49)	99.77 (19.25)	0.24
Alanine aminotransferase (IU/L)	30.57 (25.84)	29.74 (23.36)	34.3 (38.51)	28.35 (18.88)	30.97 (25.41)	0.07
Aspartate aminotransferase (IU/L)	26.91 (17.72)	25.28 (15.71)	27.58 (24.52)	26.55 (18.25)	27.75 (16.23)	0.12
HSI score	33.76 (5.94)	33.81 (6.16)	34.26 (6.84)	33.32 (5.6)	33.8 (5.7)	0.04
LAP score	44.86 (49.5)	38.83 (49.5)	39.66 (41.92)	44.77 (45.53)	49.45 (52.47)	0.04
NLFS score	−0.8 (0.86)	−0.88 (0.75)	−0.8 (1.13)	−0.79 (0.87)	−0.76 (0.82)	0.08
Alcohol consumption	3.22 (0.69)	2.97 (0.88)	2.94 (0.9)	3.3 (0.6)	3.4 (0.44)	<0.001
SMOKING						<0.001
Non-smoker	1395 (0.27)	411 (0.34)	270 (0.44)	239 (0.24)	475 (0.21)	
Ex-smoker	1900 (0.37)	466 (0.38)	190 (0.31)	369 (0.37)	875 (0.38)	
Current smoker	1839 (0.36)	338 (0.28)	152 (0.25)	389 (0.39)	960 (0.42)	
EXERCISE						0.03
Non-exercise	4248 (0.83)	1027 (0.85)	531 (0.87)	789 (0.79)	1901 (0.82)	
Irregular exercise	506 (0.1)	129 (0.11)	48 (0.08)	106 (0.11)	223 (0.1)	
Regular exercise	380 (0.07)	59 (0.04)	33 (0.05)	102 (0.1)	186 (0.08)	

Data are presented in mean (SD) unless otherwise specified.

**Table 2 nutrients-15-03901-t002:** Odds ratios and 95% confidence intervals for metabolic dysfunction-associated steatotic liver disease, cut-offed by LAP, according to alcohol consumption among non-flushers and flushers.

Alcohol Consumption Level	Non–Flusher		Flusher		*p* for Interaction
	aOR (95% CI)	*p*	aOR (95% CI)	*p*	
Age adjusted					
Non-or-Infrequent Drinker	1 (Ref.)	0.020	1 (Ref.)	<0.001	<0.001
Light-to-Heavy Drinker	1.22 (1.03–1.44)		1.47 (1.29–1.68)		
Multivariable adjusted *					
Non-or-Infrequent Drinker	1 (Ref.)	<0.001	1 (Ref.)	<0.001	<0.001
Light-to-Heavy Drinker	1.90 (1.51–2.40)		2.35 (1.94–2.84)		

Abbreviation: aOR = adjusted odds ratio, CI = confidence interval; * adjusted for age, smoking, exercise, body mass index, systolic blood pressure, total cholesterol, and fasting plasma glucose.

**Table 3 nutrients-15-03901-t003:** Odds ratios and 95% confidence intervals for metabolic dysfunction-associated steatotic liver disease, cut-offed by HSI, according to alcohol consumption among non-flushers and flushers.

Alcohol Consumption Level	Non–Flusher		Flusher		*p* for Interaction
	aOR (95% CI)	*p*	aOR (95% CI)	*p*	
Age adjusted					
Non-or-Infrequent Drinker	1 (Ref.)	0.003	1 (Ref.)	<0.001	0.156
Light-to-Heavy Drinker	1.06 (1.02–1.10)		1.07 (1.04–1.10)		
Multivariable adjusted *					
Non-or-Infrequent Drinker	1 (Ref.)	0.011	1 (Ref.)	<0.001	0.118
Light-to-Heavy Drinker	1.05 (1.01–1.09)		1.07 (1.04–1.11)		

Abbreviation: aOR = adjusted odds ratio, CI = confidence interval; * adjusted for age, smoking, exercise, body mass index, systolic blood pressure, total cholesterol, and fasting plasma glucose.

**Table 4 nutrients-15-03901-t004:** Odds ratios and 95% confidence intervals for metabolic dysfunction-associated steatotic liver disease, cut-offed by NLFS, according to alcohol consumption among non-flushers and flushers.

Alcohol Consumption Level	Non–Flusher		Flusher		*p* for Interaction
	aOR (95% CI)	*p*	aOR (95% CI)	*p*	
Age adjusted					
Non-or-Infrequent Drinker	1 (Ref.)	0.003	1 (Ref.)	<0.001	0.111
Light-to-Heavy Drinker	1.05 (1.02–1.09)		1.05 (1.02–1.08)		
Multivariable adjusted *					
Non-or-Infrequent Drinker	1 (Ref.)	0.011	1 (Ref.)	0.006	0.102
Light-to-Heavy Drinker	1.04 (1.01–1.08)		1.03 (1.01–1.07)		

Abbreviation: aOR = adjusted odds ratio, CI = confidence interval; * adjusted for age, smoking, exercise, body mass index, systolic blood pressure, total cholesterol, and fasting plasma glucose.

## Data Availability

Not applicable.

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
