# Peer review of "Association between Alcohol Consumption and Metabolic Dysfunction-Associated Steatotic Liver Disease Based on Alcohol Flushing Response in Men: The Korea National Health and Nutrition Examination Survey 2019–2021"

_nutrients, 2023, doi:10.3390/nu15183901_

Round 1
Reviewer 1 Report
In the present manuscript, the relationship between MASLD (MetALD?) and alcohol flushing response is investigated. However, the diagnosis of MASLD is based only on the LAP score, here other markers (ALT etc) should be included. Both alcohol intake and alcohol flushing response are based on simple questionnaires (more than once a month, once a month, etc...). In addition, the study cohort was not screened for existing genetic variations in alcohol-metabolizing enzymes. These limitations in the reviewers opinion significantly diminish the validity of the present study. Below you will hopefully find some helpful comments:
Line 11: sentence seems to be incomplete
Since June 2023 there is a new fatty liver disease nomenclature (PMID: 37364790), manuscript should be updated accordingly.
Have genetic variations in alcohol-metabolizing enzymes been checked in this study?
Authors should also discuss the possible mechanism in more detail why alcohol flushing syndrome affects MASLD development.
LAP is not enough to diagnose NAFLD, as no ultrasound data seems to be available it should at least be combined with ALT and AST
Is there a difference in alcohol intake between flusher and non-flusher?
Group of non-or-infrequent drinkers should also be divided in flusher and non-flusher (at least infrequent drinkers should know?)
Moderate editing of English language required
Reviewer 2 Report
The present study investigated the association between alcohol consumption and NAFLD in Korean men, considering their alcohol flushing response and utilizing the lipid accumulation product (LAP) score. The study emphasize the importance of alcohol flushing as a potential indicator of NAFLD risk in Korean men and highlight the need for further research to understand the underlying mechanisms and develop targeted preventive strategies. The study was overall well conducted. The major concern is that the diagnosis criteria of NAFLD was not universally recognized and should be further discussed.
Moderate editing of English language required
Round 2
Reviewer 1 Report
Authors addressed the reviewers comments. However, I have one minor comment. I recommend reading PMID: 37364790 more carefully (Rinella et al., 2023). NAFLD is now referred to as MASLD, not MAFLD. Quote from the study mentioned above: "The name chosen to replace NAFLD was metabolic dysfunction-associated steatotic liver disease (MASLD)"
Minor editing of English language required
